# Effectiveness of physical and mental health interventions for young people with heart conditions: protocol for a systematic review and meta-analysis

Lora Capobianco ,[1,2] Joy Adewusi,[1] Beth Cooper,[1] Andrew Belcher,[1] Adrian Wells[2]

¹Research and Innovation, Greater Manchester Mental Health NHS Foundation Trust, Manchester, UK
²School of Psychological Sciences, Faculty of Biology, Medicine and Health, The University of Manchester, Manchester, UK

**Correspondence to**
Dr Lora Capobianco;
Lora.Capobianco@gmmh.nhs.uk

## ABSTRACT

**Introduction** Cardiovascular disease is among the most common of non-communicable diseases, affecting 13.9 million children and young people (CYP) globally. Survival rates for CYP with heart conditions are rising, however, support for adjusting to life with a heart condition is lacking, as such it is unsurprising that one in three suffer from anxiety, depression or adjustment disorder. The proposed review aims to identify and assess the effectiveness of physical and mental health interventions across physical and mental health outcomes in young people with cardiac conditions using narrative synthesis and meta-analysis if appropriate.

**Methods and analysis** Embase, Medline, PubMed, PsycINFO, Cochrane Databases, Web of Science and reference lists of relevant publications will be searched from 1980 to June 2022 for articles published in English or Italian. Screening, data extraction, intervention coding and risk of bias will be performed by two independent reviewers using an extraction checklist. Intervention content and features will be identified and reported using the Template for Intervention Description and Replication checklist. A narrative review of the included studies will be conducted. If possible and appropriate, a random-effects model meta-analysis will be conducted to calculate the pooled within-group and between-group effect sizes for the primary outcome measures. If sufficient data are available, a subgroup meta-analysis will investigate whether specific intervention types are associated with different levels of intervention effectiveness.

**Ethics and dissemination** This systematic review does not directly involve the use of human beings, therefore, there is no requirement for ethical approval. Findings will be disseminated through peer-reviewed publication and in various media, such as conferences, congresses or symposia.

**PROSPERO registration number** CRD42022330582.

## INTRODUCTION

Childhood heart conditions, such as congenital heart disease (CHD), cardiomyopathy and arrythmias, are among the most common childhood non-communicable diseases, impacting 13.8 million children and young people (CYP) under the age of 20 globally.[1]

## STRENGTHS AND LIMITATIONS OF THIS STUDY

⇒ The review addresses a gap in the literature by being, to our knowledge, the first to explore the effectiveness of mental and physical intervention on mental and physical outcomes in young people with heart conditions, thereby overcoming a limitation of previous reviews.

⇒ The review is not restricted to randomised trials but will incorporate a range of study designs including non-randomised interventional studies, pilot studies, uncontrolled trials and randomised trials.

⇒ While the study aims to evaluate the magnitude of the effect of current interventions, understanding of the mechanisms of change will not be evaluated but is an important area for future studies.

⇒ While the review aims to include a broad age range, studies of young adults transitioning from paediatric to adult services will be excluded.

Survivors of childhood heart conditions face challenges and comorbidities which impact their ability to function,[2] future employment and progression into independent adulthood.[3 4] CYP with heart conditions are also less likely to engage in physical activity in comparison with healthy controls,[5] which increases their risk for atherosclerosis, cardiovascular disease, obesity and diabetes.[6]

Recommendations from the American Heart Association and European Association of Cardiovascular Prevention and Rehabilitation[7] have noted the importance of CYP engaging in physical activity, exercise and maintaining a healthy lifestyle (ie, healthy diet, engaging in exercise, reduced smoking and alcohol intake, and medication adherence). Regular moderate to vigorous physical activity in CYP has been shown to improve both physical and mental health.[8] This is of particular importance as CYP with heart conditions have a reduced quality of life[9–11] and poorer psychological functioning, with

30% suffering from anxiety or depression[10 12–15] and 41% experience psychological maladjustment.[10]

Interventions to improve outcomes in CYP with CHD have predominantly focused on improving exercise capacity or engagement in physical activity.[16] van Deutekom and Lewandowski[16] conducted a narrative review of physical activity modification in CYP with CHD. Eight interventions were included that aimed to improve physical activity. The authors noted the heterogeneity in the interventions offered, with four of the trials reporting a significant increase in physical activity levels following the intervention. These results are in line with previous systematic reviews which have also noted that physical activity and exercise training programmes significantly improve physical health outcomes (ie, peak $VO_2$, activity levels and muscle strength).[17]

While systematic reviews have noted that exercise and physical activity training improve physical health outcomes, only one review[18] has commented on improvements in quality of life, with one review conducted to evaluate mental health interventions for CYP with CHD.[19] Tesson et al[19] conducted a systematic review of psychological interventions for childhood-onset heart disease and found only two studies focused on interventions for CYP.[20 21] The results were disappointing, and at post-treatment they did not decrease anxiety, depression[20] or improve quality of life.[21]

Previous research has predominantly focused on synthesising the evidence on exercise interventions in improving physical health outcomes, or mental health interventions in improving mental health outcomes in CYP with heart conditions. Reviews have not evaluated the effectiveness of such interventions and the size of the effect they produce. Interestingly, evidence from the wider literature (outside of cardiac health) suggests that exercise and physical activity interventions have produced significant improvements in anxiety and low mood.[22–24] Spruit et al[25] conducted a meta-analysis of 57 physical activity interventions and their effects on psychosocial outcomes including internalising and externalising problems. They found a small-to-medium effect size of physical activity interventions on reducing externalising (Cohen's d=0.320) and internalising (Cohen's d=0.316) problems in CYP. While the results are promising the effect sizes are limited and are yet to outperform the benefits of mental health interventions on improving mental health outcomes. While these results are promising similar effects are yet to be explored in CYP with CHD. Furthermore, while physical activity/exercise interventions appear to improve mental health outcomes, much less is known concerning the effect of mental health interventions on physical health outcomes.

As such, the aim of this review is to fill an important gap in the literature by addressing the following questions: (1) how effective are mental and physical health interventions in improving mental and physical health outcomes for young people with heart conditions (2) do mental health interventions improve physical health outcomes and if so, what is the size of the effect and (3) do exercise interventions improve mental health outcomes and if so what is the size of the effect.

## METHODS AND ANALYSIS

This protocol is based on the Preferred Reporting Items for Systematic Reviews and Meta-Analyses Protocols (PRISMA) guidelines (see online supplemental appendix 1) and review findings will be reported in line with PRISMA guidance.[26] The review has been registered with the International Prospective Register of Systematic Reviews (PROSPERO, registration number: CRD42022330582).

### Search strategy

The following databases will be searched: MEDLINE, EMBASE, PsycINFO, Web of Science, Cochrane Central Register of Controlled Trials (CENTRAL), DARE (via the Cochrane Library) and Science Citation Index (via Web of Science).

The search method will include extensive database searching and supplementary searching including forwards and backwards citation chasing, handsearching of any key related reviews identified during the search process and additional searching on topic specific websites. The strategy will use both controlled headings (eg, Medical Subject Headings (MeSH)) and free-text terms. Search terms will be grouped according to three concepts: heart conditions, mental health interventions and exercise programmes. Terms relating to these concepts will be developed based on the available literature. The search strategy can be seen in online supplemental appendix 2.

### Study inclusion criteria

Eligible studies will be peer-reviewed publications in English or Italian that include young people (age 6–18 years old) diagnosed with any form of CHD or an implanted cardiac device. Studies will be included if they are interventional studies designed to evaluate an exercise or physical activity intervention and/or a mental health intervention. Interventional studies will include randomised controlled trials, non-randomised interventional studies, pilot studies and uncontrolled trials. To be included, studies must evaluate an exercise or physical activity intervention or psychosocial intervention. Exercise interventions may include a structured cardiac rehabilitation programme. There are no restrictions on the setting of interventions, these can be home based, or centre based (ie, hospital or community setting). A control group is not a necessary requirement for inclusion within this review. Comparator conditions may include standard care, waiting lists, active controls, no intervention control groups. Studies published before 1980 are excluded as youth mental health was not considered a discrete field prior to 1980.[27]

### Information sources

An extensive database search will be conducted using MEDLINE, EMBASE, PsycINFO, Web of Science,

Cochrane Database of Systematic Reviews (via the Cochrane Library), Cochrane Central Register of Controlled Trials (CENTRAL), DARE (via the Cochrane Library) and Science Citation Index (via Web of Science). Databases will be searched from inception to May 2022. The citation lists of included references will be checked and forwards citation searching (identifying where included references have been cited) will be carried out using Web of Science and Google Scholar. For the current review, any included articles will be subject to forward and backwards citation searching. Study authors will be contacted where information is missing and/or the full-text article is unavailable.

### Data management
The searches will be recorded using PRISMA guidelines, including the list of databases searched, recording of the dates (original and updated) searched and the strategies used for each database. All the articles identified by the database searches will be imported into EndNote reference management software V.X20 prior to deduplication and screening. Duplicates not identified by Endnote will be manually removed. References identified by the updated search will be handled in the same manner. Articles will then be exported to Rayyan, a web-based software for the management of initial screening and selection of papers in systematic reviews.[28] Papers will then be transferred to COVIDENCE to manage full-text review and data extraction.[29]

### Study selection
Article titles and abstracts will be screened against the inclusion/exclusion criteria by two independent reviewers (JA and AB). Following this, full texts of the remaining articles will be screened against the inclusion/exclusion criteria. Any conflicts or uncertainty will be resolved by discussion with the research team. All articles deemed to be included in the review will be discussed with the research team and any discrepancies of opinion will be resolved by LC and AW. Reasons for inclusion/exclusion will be recorded and a PRISMA flow chart will be used to show the details of the selection process.

### Data extraction
One reviewer will independently extract data in each eligible study using a data extraction form. Other reviewers will check the extracted data to ensure accuracy and completeness of the data.

The following information will be extracted where available:
► General: author, publication date, country of origin, digital object identifier (DOI).
► Study characteristics: aims/hypotheses, study design, inclusion/exclusion criteria, recruitment methods.
► Sample characteristics: size, gender distribution, age, comorbidity information and baseline clinical characteristics including cardiac diagnoses.

► Intervention: duration (number and length of sessions) mode of delivery, drop-out rates, attendance statistics, follow-up times (if applicable), quality of intervention using Template for Intervention Description and Replication checklist.[25]
► Data analysis: comparison conditions (if applicable), statistical procedures used to analyse data.
► Primary and secondary outcomes.
► Key findings: pretreatment and post-treatment descriptive data, follow-up data, effect size of intervention.

Where a study is described across multiple publications, an attempt will be made to extract and combine all the available data. Study authors will be contacted if data is missing. If multiple papers are published using the same dataset only the study most relevant to the review will be included to avoid duplication of participants.

### Outcomes
The most commonly reported outcomes for physical health are cardiorespiratory fitness assessed by peak oxygen consumption and daily physical activity (subjective and objective outcomes). Primary outcomes for mental health are often quality of life, but we will consider available evidence on specific anxiety, depression, trauma and adjustment symptoms.

Additional outcomes are changes in health-related behaviours (physical activity, diet, smoking, alcohol, sedentary behaviour and medication adherence) as interventions aimed at heart conditions would likely aim to improve modifiable cardiovascular disease risk factors. Additional outcomes will include psychological well-being, lipid profile, blood pressure, weight/body mass index, number and type of cardiac events, rehospitalisation and mortality.

### Quality assessment/risk of bias in individual studies
Quality of individual papers will be assessed using an appropriate Critical Appraisal tool based on study design as described in the manual for developing National Institute for Health and Care Excellence (NICE) guidelines,[30] such as the Risk of Bias tool developed by The Cochrane Collaboration (version 2).[31] One reviewer (JA) will complete the assessment and a second reviewer (BC) will check a random 20% of the studies. Intraclass correlation coefficient will be used to indicate the level of agreement between the reviewer assessments, in a two-way random-effects model.

### Data synthesis
A narrative review of the findings will be conducted. This will comprise information summarising study characteristics, patient characteristics, information pertaining to the delivery of exercise interventions (eg, type of exercise, number of sessions, length of sessions, number of attendees), outcome variables, follow-up, quality assessment, treatment effects (eg, within-and between-group effect sizes). These will also be reported in tabular form.

If possible and appropriate, a random-effects model meta-analysis will be conducted to calculate the pooled within and between-group effect sizes. Between-group effect sizes will be calculated for the primary outcome measure in the treatment arm (exercise/mental health intervention) compared with waitlist control or active treatment control arms (at end of treatment and at longest follow-up time available), respectively; these will be analysed separately. Heterogeneity will be calculated using the I² statistic, where 25%, 50%, 75% are interpreted as referring to low, moderate and high levels of heterogeneity.[32] Discussions of suitability of inclusion of studies in any meta-analysis will be held with the research team. The meta-analysis will be conducted using Review Manager V.5 (RevMan), according to the Cochrane guidelines.[33]

For continuous variables, the mean difference will be calculated if the same measurement scale was used, alternatively the standardised mean difference will be calculated (with 95% CI). For dichotomous variables, proportions will be compared using risk ratios (with 95% CIs). If a meta-analysis is not possible, data will be summarised using a narrative synthesis approach.

Subgroup analyses will be conducted where appropriate and possible. This might consist of subgroups of populations (eg, type of CHD, age) or intervention characteristics (home vs centre based, aerobic vs resistance exercise, presence or not of psychological treatment elements)

### Meta-bias(es)
Where there are a minimum of 10 studies available which allow for any between-group effect size calculations, publication bias will be investigated using visual inspection of funnel plots (looking for asymmetry indicative of publication bias).

### Confidence in cumulative evidence
If appropriate and possible, the Grading of Recommendations Assessment, Development and Evaluation guidelines approach will be used to make a judgement about the overall quality of the evidence base.

### Patient and public involvement
Patients and the public will not be involved directly in the design and conduct of the review. However, the development of our review question was informed by a patient and public involvement group with CYP with heart conditions. The results of these interviews highlighted misconceptions regarding physical activity and exercise and pointed to unhelpful coping styles that could have significant psychosocial impact.

### Ethics and dissemination
Ethical approval is not required for this systematic review because primary data will not be collected. This systematic review protocol is registered in the International Prospective Register of Systematic Reviews (http://www.crd.york.ac.uk/PROSPERO). The result of the review will be disseminated through publication in an academic journal and disseminated as part of future work in the development of exercise-based interventions for CYP with CHD.

## DISCUSSION
Interventions that include physical and mental health support for CYP with heart conditions have the potential to inform future research and improve patient outcomes. The findings could contribute to overcoming barriers and misconceptions associated with exercise and physical activity thereby promoting healthier lifestyles in CYP with heart conditions.

The review will identify exercise and physical activity interventions, and mental-health interventions that are associated with change in important health outcomes in CYP with heart conditions. The results will provide key insights into the content and design of successful interventions for this patient group and help to inform the design of future exercise-based and integrative psychosocial interventions for CYP with CHD.

While the review aims to include a broad age range this may exclude young adults who are transitioning between paediatric and adult services and may exclude a population who may benefit from mental health and exercise interventions, as such future reviews should aim to evaluate those transitioning between services. In addition, while the aim of the review is to assess the nature and effectiveness of the interventions, less is likely to be known on the mechanisms of change that may lead to greater improvements in outcomes. As such, future reviews should aim to explore the mechanisms of change within interventions.

This review addresses a gap in the literature by evaluating effectiveness of exercise and mental-health interventions in this young patient group on both mental and physical health outcomes. This systematic review is being conducted in the broader context of developing a cardiac rehabilitation programme that includes physical and mental health support for CYP with heart conditions.

**Contributors** LC and AW conceived the study,and wrote the initial grant protocol. All authors contributed to the first and subsequent drafts of the manuscript and approved the final version for submission. JA contributed to the protocol methods, first and subsequent drafts of the protocol. BC and AB contributed to editing the manuscript.

**Funding** This study is funded by the National Institute for Health Research (grant number NIHR203634).

**Disclaimer** The views expressed are those of the author(s) and not necessarily those of the NIHR or the Department of Health and Social Care.

**Competing interests** None declared.

**Patient and public involvement** Patients and/or the public were involved in the design, or conduct, or reporting, or dissemination plans of this research. Refer to the Methods section for further details.

**Patient consent for publication** Not applicable.

**Provenance and peer review** Not commissioned; externally peer reviewed.

**ORCID iD**
Lora Capobianco http://orcid.org/0000-0001-6877-8650

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
