## [Reviewer comments · BMJ Open]

ARTICLE DETAILS

TITLE (PROVISIONAL)	Effectiveness of physical and mental health interventions for young people with heart conditions: protocol for a systematic review and meta-analysis
AUTHORS	Capobianco, Lora; Adewusi, Joy; Cooper, Beth; Belcher, Andrew; Wells, Adrian

VERSION 1 – REVIEW

REVIEWER	Tanya Hauck Centre for Addiction and Mental Health, Addictions
REVIEW RETURNED	04-Oct-2022

GENERAL COMMENTS	This protocol by Capobianco et al describes a systematic review and meta-analysis of mental and physical health interventions for young people with heart conditions. The authors provide a thoughtful background and rationale for the protocol. They carefully detail their approach and analysis plan. I find the primary hypothesis very interesting and important, specifically that mental health interventions improve physical health outcomes, and that exercise interventions improve mental health outcomes. I would recommend elaborating on this. For example, why do you propose this is the case? Is there synergy between these interventions? What is the proposed mechanism by which this would occur? It is important to perhaps cite some papers that identify this linkage, and also provide evidence that there will be enough data to support the overall meta-analysis. For example, papers showing that mental health interventions improve physical health. Please discuss any ethics approval related to the focus groups held with CYP, e.g. were these for a different study that had ethics approval? Finally, please be specific on page 5, line 25 about what is meant by “maintaining a healthy lifestyle”, although it is described later (“(physical activity, diet, smoking, alcohol, sedentary behaviour, and medication adherence”).
--

REVIEWER	Elias Mpofu Sydney University, Rehabilitation
REVIEW RETURNED	17-Oct-2022

GENERAL COMMENTS	The aims of the protocol are laudable. However, authors need to reconsider their claim that there have been no review in physical and mental health intervention outcomes for children and youth with heart disease. This claim is way too spacious (and as a matter of fact they cite a study by Tesson e tal.). Authors are better of seeking to review the evidence for specific interventions types and the
--

	contexts for those interventions. A related concern is with the authors being overly inclusive of the study types to include their review: benchmarking, randomized, non-randomized, pilot, uncontrolled etc. This inclusiveness compromises the interpretability of findings. Authors then plan to use the PRISMA guidelines, when with study types for which PRISMA would be inappropriate. There are other guidelines for the variety of study types they envisage, including CONSORT, STROBE, GRADE et cetera. These major limitations of the protocol suggest a need for a comprehensive rewrite for a more focused, feasible study plan that is the case this submission.
--	--

VERSION 1 – AUTHOR RESPONSE

Reviewer: 1

Dr. Tanya Hauck, Centre for Addiction and Mental Health

Comments to the Author:

This protocol by Capobianco et al describes a systematic review and meta-analysis of mental and physical health interventions for young people with heart conditions. The authors provide a thoughtful background and rationale for the protocol. They carefully detail their approach and analysis plan.

I find the primary hypothesis very interesting and important, specifically that mental health interventions improve physical health outcomes, and that exercise interventions improve mental health outcomes. I would recommend elaborating on this. For example, why do you propose this is the case? Is there synergy between these interventions? What is the proposed mechanism by which this would occur? It is important to perhaps cite some papers that identify this linkage, and also provide evidence that there will be enough data to support the overall meta-analysis. For example, papers showing that mental health interventions improve physical health.

Response: The reviewer raises important questions, and we suspect some of these will be outside the scope of the available data to permit extensive analysis (i.e. mechanisms). However, we have added some papers citing the links between physical activity and mental health and how physical activity may improve mental health outcomes. Less is known concerning the effects of mental health interventions on improving physical health outcomes. The ability for us to assess mechanisms depends in part on the data available in current reviews and we agree with the reviewer that this is an important key question that is worthy of further research and have noted the extent of available data as a limitation.

Please discuss any ethics approval related to the focus groups held with CYP, e.g. were these for a different study that had ethics approval?

Response: This did not require ethical approval as this was completed as part of our regular patient and public involvement group meetings. We have revised the use of 'focus groups' as a term to avoid confusion.

Finally, please be specific on page 5, line 25 about what is meant by “maintaining a healthy lifestyle”, although it is described later (“(physical activity, diet, smoking, alcohol, sedentary behaviour, and medication adherence”).

Response: Thank you, we have added further details on page 4, we now state that maintaining a healthy lifestyle includes a healthy diet, engaging in exercise, medication adherence and reduced smoking and alcohol.

Reviewer: 2

Dr. Elias Mpofo, Sydney University, University of North Texas

Comments to the Author:

The aims of the protocol are laudable. However, authors need to reconsider their claim that there have been no review in physical and mental health intervention outcomes for children and youth with heart disease. This claim is way too spacious (and as a matter of fact they cite a study by Tesson et al.). Authors are better off seeking to review the evidence for specific intervention types and the contexts for those interventions.

Response: We did not intend to make the claim that there had been no review in physical and mental health intervention outcomes, but rather we were referring to the lack of reviews that evaluate the effectiveness of the interventions. While reviews have conducted narrative syntheses of the literature they had not to our knowledge conducted a meta-analysis on the data to evaluate the size of effects as specifically proposed within the review. We have however removed this from the paper in order to avoid any confusion

A related concern is with the authors being overly inclusive of the study types to include their review: benchmarking, randomized, non-randomized, pilot, uncontrolled etc. This inclusiveness compromises the interpretability of findings.

Response: We disagree that this compromises the review, but instead is a strength and the results can be analyzed as controlled vs uncontrolled designs. We have however modified the number of designs included to remove case studies and case reports.

Authors then plan to use the PRISMA guidelines, when with study types for which PRISMA would be inappropriate. There are other guidelines for the variety of study types they envisage, including CONSORT, STROBE, GRADE et cetera.

Response: We considered PRISMA because it is an evidence-based minimum set of items for reporting in systematic reviews and meta-analyses that primarily focuses on evaluating the effects of interventions. As our study is reviewing interventions PRISMA is appropriate. However, we also note that we will use the TIDIER checklist to describe interventions and as stated on page 11, If appropriate and possible, the Grading of Recommendations Assessment, Development, and

Evaluation guidelines (GRADE) approach will be used to make a judgement about the overall quality of the evidence base.

These major limitations of the protocol suggest a need for a comprehensive rewrite for a more focused, feasible study plan that is the case this submission.

Response: Thank you for raising these concerns, I hope we have been able to allay them by providing the additional information in answering the reviewers questions. We believe the study is feasible as set out, and that for any review of this kind, with a potentially limited range of papers that can address our questions, it is prudent not to be too tightly focused.

VERSION 2 – REVIEW

REVIEWER	Tanya Hauck Centre for Addiction and Mental Health, Addictions
REVIEW RETURNED	28-Nov-2022

GENERAL COMMENTS	Thank you for the opportunity to review the revisions of this paper. Overall, the authors have addressed the concerns raised. I have no major concerns. I think for the ultimate paper (but perhaps not for the protocol) it might be helpful to see a table of how mental/physical interventions lead to mental/physical outcomes – like a two-by-two table. Please note, page 8, line 1, it should be future tense, not past tense.
--

REVIEWER	Elias Mpofu Sydney University, Rehabilitation
REVIEW RETURNED	26-Nov-2022

GENERAL COMMENTS	Authors are way too inclusive of the type of studies are no rationale why they cannot restrict to a certain study types for manageability and interpretability of findings from the systematic and meta-analysis they propose. Effectiveness studies are not the same as exploratory or proof of concept studies where, perhaps there may be need to be inclusive for any trends in the evidence across method approaches. Effectiveness studies typically follow efficacy studies, and would be largely quantitative. The authors do not acknowledge the limitation their protocol seeking to do all about everything. This is not a trivial concern should they consider to continue with their very broadly construed review.
--

VERSION 2 – AUTHOR RESPONSE

Reviewer: 2

Authors are way too inclusive of the type of studies are no rationale why they cannot restrict to a certain study types for manageability and interpretability of findings from the systematic and meta-analysis they propose. Effectiveness studies are not the same as exploratory or proof of concept studies where, perhaps there may be need to be inclusive for any trends in the evidence across method approaches. Effectiveness studies typically follow efficacy studies, and would be largely quantitative. The authors do not acknowledge the limitation their protocol seeking to do all about

everything. This is not a trivial concern should they consider to continue with their very broadly construed review.

Response: Restricting the number of study types to include only controlled studies or only uncontrolled designs would run the risk of limiting the conclusions we might draw from the literature. At this early stage of evidence synthesis it is safer to be over rather than under-inclusive, with the intention of partitioning the analysis dependent on study design. We do not intend to combine uncontrolled and controlled designs in our analyses as perhaps suggested by the reviewer but we would separate these in our analysis, interpretation and understanding. We believe it strengthens our review that we are able to consider both controlled and uncontrolled designs and discuss current findings, furthermore we might find informative contrasts in study effects associated with different study designs.

Reviewer: 1

Thank you for the opportunity to review the revisions of this paper.

Overall, the authors have addressed the concerns raised. I have no major concerns. I think for the ultimate paper (but perhaps not for the protocol) it might be helpful to see a table of how mental/physical interventions lead to mental/physical outcomes – like a two-by-two table.

Please note, page 8, line 1, it should be future tense, not past tense.

Response: Thank you for your suggestion on a table for a future paper, we will endeavor to undertake this when publishing the results of the review.

We have revised the typo on page 8 line 1 to be future tense rather than past.